# Iterative Image Inpainting with Structural Similarity Mask for Anomaly Detection

## Abstract

Autoencoders have emerged as popular methods for unsupervised anomaly detection. Autoencoders trained on the normal data are expected to reconstruct only the normal features, allowing anomaly detection by thresholding reconstruction errors. However, in practice, autoencoders fail to model small detail and yield blurry reconstructions, which makes anomaly detection challenging. Moreover, there is objective mismatching that models are trained to minimize total reconstruction errors while expecting a small deviation on normal pixels and a large deviation on anomalous pixels. To tackle these two issues, we propose the iterative image inpainting method that reconstructs partial regions in an adaptive inpainting mask matrix. This method constructs inpainting masks from the anomaly score of structural similarity. Overlaying inpainting mask on images, each pixel is bypassed or reconstructed based on the anomaly score, enhancing reconstruction quality. The iterative update of inpainted images and masks by turns purifies the anomaly score directly and follows the expected objective at test time. We evaluated the proposed method using the MVTec Anomaly Detection dataset. Our method outperformed previous state-of-the-art in several categories and showed remarkable improvement in high-frequency textures.

## 1 Introduction

Anomaly detection (AD) is the identification task of the rarely happened events or items that differ from the majority of the data. In the real world, there are many applications, such as the medial diagnosis (Baur et al., 2018; Zimmerer et al., 2019a), defect detection in the factories (Matsubara et al., 2018; Bergmann et al., 2019), early detection of plant disease (Wang et al., 2019), and X-Ray security detection in public space (Griffin et al., 2018). Because manual inspection by humans is slow, expensive, and error-prone, automating visual inspection is the popular application of artificial intelligence. In transferring knowledge from humans to machines, there is a lack of anomalous samples due to their low event rate and difficulty annotating and categorizing various anomalous defects beforehand. Therefore, AD methods typically take unsupervised approaches that try to learn compact features of data from normal samples and detect anomalies by thresholding anomaly score to measure the deviation from learned features. To deal with high-dimensional images and learn their features, it is popular to use deep neural networks (Goodfellow et al., 2016).

In this work, we focus on the reconstruction-based unsupervised AD. This attempts to reconstruct only the normal dataset and classify the normal or anomalous data on thresholding reconstruction errors (An & Cho, 2015). The architectures are based on deep neural networks such as deep autoencoders (Hinton & Salakhutdinov, 2006), variational autoencoders (VAEs) (Kingma & Welling, 2013; Rezende et al., 2014), or autoencoders with generative adversarial networks (GANs) (Goodfellow et al., 2014). These models compress the high-dimensional information into the data manifold in lower-dimensional latent space by reconstructing input data under certain constraints for latent space, such as a prior distribution or an information bottleneck (Alemi et al., 2016).

The reconstruction-based AD approach issue is that autoencoders fail to model small details and yield blurry image reconstruction. This is especially the case for the high-frequency textures, such as carpet, leather, and tile (Bergmann et al., 2019). Dehaene et al. (2020) also pointed out that there is no guarantee of the generalization of their behavior for out-of-samples, and local defects added to normal images could deteriorate whole images. In the viewpoint of the signal-to-noise ratio (SNR),

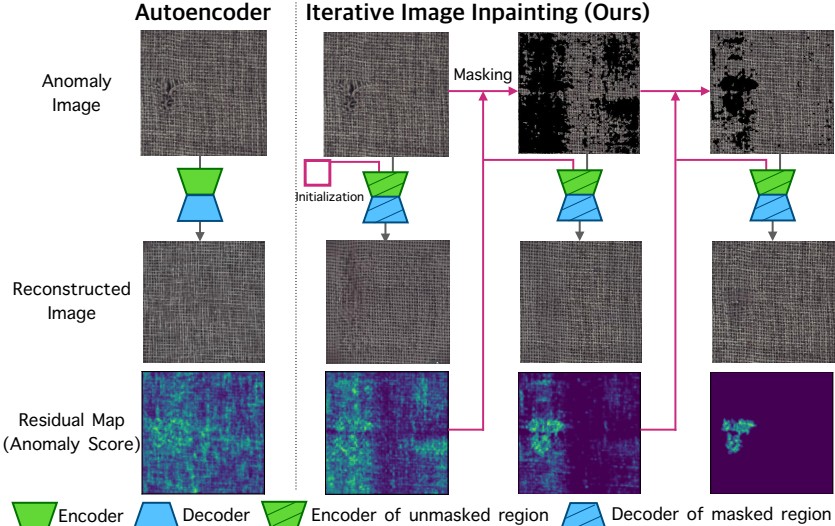

Figure 1: Autoencoder vs. I3AD method (Ours). Autoencoder fails to reconstruct the high-frequency texture of "carpet" and yields a large residual map. Ours utilizes the residual map as an inpainting mask and iteratively updates the reconstructed image and the residual map.

it is interpreted that blurry reconstruction makes anomaly signals (reconstruction errors in anomaly pixels) unclear and increases normal noises (reconstruction errors in normal pixels). Since the SNR explains the feasibility of AD by thresholding a sample-wise reconstruction error, the low SNR makes AD challenges.

We point out an additional issue about the gap between optimized function at training and evaluated function at testing. Rethinking our goal in unsupervised AD, it is concluded not to minimize reconstruction errors merely but to maximize the SNR. Although models are trained to minimize a sample-wise reconstruction error, they are expected to have a large deviation of anomaly pixels and a small deviation of normal pixels at testing.

In this paper, we propose *I3AD* (Iterative Image Inpainting for Anomaly Detection). As Figure 1, our method utilizes an inpainting model that only encode unmasked regions and reconstruct masked regions instead of vanilla autoencoders. Once computed reconstruction errors, it is recycled as an inpainting mask for the next iteration. We show that the iterative update enhances the reconstruction quality and satisfies the expected objective to maximize the expected SNR at testing. Through experiments and analysis on the MVTecAD dataset shows that our I3AD outperforms existing methods on nine categories and has ave. $+11.6\%$ improvement on texture category.

## 2 METHODOLOGY

### 2.1 HIGH-LEVEL IDEA

We think of unsupervised AD using autoencoders. Here, we implicitly assume anomalies show up in partial regions, and pixels in surrounding regions obey by the distribution of normal datasets. Therefore, the sample-wise anomaly score based on reconstruction errors is the summation of two types of pixels: (1) pixels of normal regions (normal noises) and (2) pixels of anomalous regions (anomaly signals). We expect an ideal model with zero errors on normal regions and distinguishable per-pixel scores on anomalous regions, leading to the high SNR.

Inheriting vanilla autoencoders' architecture does not help to resolve the low SNR issue mentioned by Bergmann et al. (2019) and Dehaene et al. (2020). Indeed, autoencoders are forced to encode whole images with anomalous pixels of local defects and attempt to decode whole images with background normal pixels of fine structures. They do not learn their behavior to encode unseen anomalous pixels. That anomalous information can affect the whole image decoding.

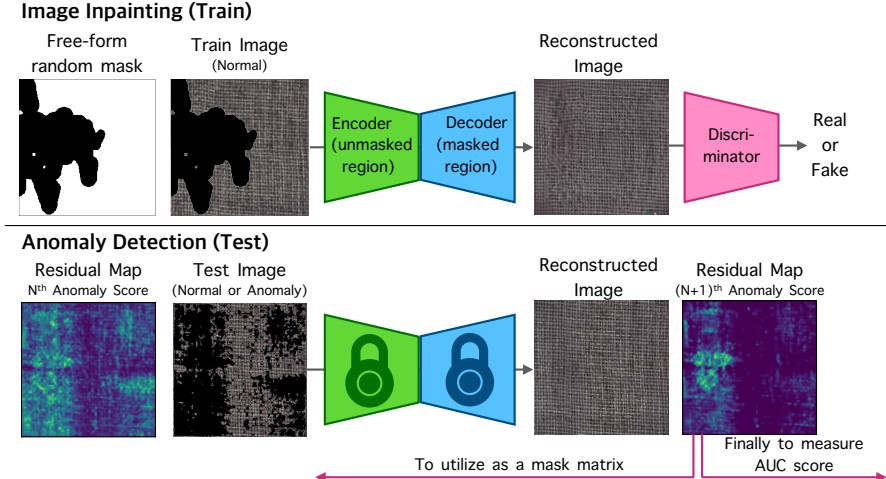

Figure 2: Model overview of iterative image inpainting method for AD (I3AD). At the training step, we solve a general image inpainting task in normal datasets. At the testing step, a generator receives test images with an adaptive mask and only reconstructs masked regions. This adaptive mask is updated from previous reconstruction results during iterations.

One approach to resolve the issue is a combination of the per-pixel identity function and conditional autoencoder. Compared to vanilla autoencoders, conditional autoencoders can encode only normal regions and decode only anomalous regions. The per-pixel identity function could copy the remaining unreconstructed regions. This model architecture is the same as an image inpainting model. Actually, deep inpainting models are designed by conditional autoencoders that encode unmasked regions and fill in masked regions under a certain mask matrix (Yu et al., 2019). However, the image inpainting method for AD falls into the tautology trap that we do not know a perfect inpainting mask matrix in advance, and detecting anomalous regions is the main goal to achieve.

The key ideas to disentangle this tautology are mask generation by the anomaly score and iterative update of a mask matrix. Updating the inpainting mask matrix dynamically controls encoding and decoding information balances with a pixel-wise confidence level of anomaly scores. A generator gradually receives more information on potentially normal pixels and focuses on the suspected pixels during iterations. Furthermore, this process not only reduces background noises but also improves the SNR directly.

## 2.2 ITERATIVE IMAGE INPAINTING FOR ANOMALY DETECTION

Following the above discussion, we construct our I3AD method by an inpainting generator and a mask generation module. As a mask generation module, we explain the detail in the next subsection.

Our model overview is depicted in Figure 2. We construct an inpainting generator using conditional generative adversarial networks (cGANs) (Isola et al., 2017) and train its networks by a general image inpainting task under a normal dataset. We feed normal images partially hidden by randomly generated masks into generator networks and train them to decode the masked pixels from the unmasked pixels and corresponding bool mask matrix. A discriminator network distinguishes generated images from normal images. A generator is rewarded for fooling a discriminator, while a discriminator is rewarded for detecting the generated images, respectively. This training can be considered as the two-player min-max game that a generator and a discriminator compete. As a result, the inpainting model tries to find the optimal point on the loss function as below:

$$\min_G \max_D \ E_{x \sim P_{\text{data}}(x)}[\log D(x)] + E_{x \sim P_{\text{data}}(x), M \sim P(M)}[\log(1 - D(G(\hat{x} = M \odot x, M)))],$$

where $x$ and $\hat{x}$ are real samples from data distribution $P_{\text{data}}(x)$ and their masked real samples. $\odot$ is an element-wise product. $M$ is the corresponding bool mask matrix generated from the random distribution $P(M)$. $G(\hat{x}, M)$ is an image inpainting network that takes an incomplete image and masked matrix. $D(x)$ denotes a binary classifier whether an image is generated or real.

We borrow and customize Spectral-Normalized Markovian GAN (SN-PatchGAN) architecture following by Yu et al. (2019). The network consists of two networks: a coarse-to-fine generator network with attention module and gated convolution, and a spectral normalized Markovian (Patch) discriminator network. Our I3AD expects to handle more fine structured masks than usual free-form masks. Therefore, to better handle more irregular masks, we apply the self-attention module (Zhang et al., 2018) instead of the contextual attention module originally designed for large rectangular masks as described in Yu et al. (2018; 2019).

To stabilize the training of GAN, we adapted proposed spectral normalization (Miyato et al., 2018) for discriminator's layers. As approximation of objective min-max loss function (Miyato et al., 2018), we also derived loss functions respectively for generator $L_G$ and discriminator $L_D$ below:

$$L_G = -E_{x \sim P_{\text{data}}(x), M \sim P(M)}[D^{\text{sn}}(G(\hat{x} = M \odot x, M))]$$
$$L_D = E_{x \sim P_{\text{data}}(x)}[\text{ReLU}(1 - D^{\text{sn}}(x))]$$
$$+ E_{x \sim P_{\text{data}}(x), M \sim P(M)}[\text{ReLU}(1 + D^{\text{sn}}(G(\hat{x} = M \odot x, M)))],$$

where $D^{\text{sn}}(x)$ denotes spectral-normalized discriminator. ReLU is the abbreviation of Rectified Linear Units activation function, defined by $\text{ReLU}(x) = \max(0, x)$. For a generator network, we use a spatially discounted $l_1$ reconstruction loss (Yu et al., 2018).

At the test step, we fix all trainable parameters in the I3AD generator. The I3AD generator receives test images that are normal or anomalous images and adaptive mask matrices. Mask matrices are constructed from the pixel-wise reconstruction errors between original images and generated images at the previous iteration step. Since mask matrices are dynamically updated and shrunken during iterations, the I3AD generator generates masked regions intensively, leveraging the gradually increasing information of the surrounding unmasked regions.

### 2.3 INPAINTING MASK OF STRUCTURAL SIMILARITY (SSIM MASK)

In anomaly segmentation tasks, structural similarity measure (SSIM) index (Wang et al., 2004) sharply measures the small anomalous change (Bergmann et al., 2018). Details of SSIM calculation is in Appendix A.

We propose the structural similarity mask (SSIM-Mask) to mask anomalous pixels during test iterations. SSIM-Mask $M_i$ is a binary mask thresholding the pixel-wise SSIM Index between an input image $x_0$ and the reconstructed image $\tilde{x}_i$ at $i^{\text{th}}$ iteration step, defined as

$$M_i = \begin{cases} 1 & \text{if } a_i(x) \geq u \\ 0 & \text{otherwise,} \end{cases}$$
$$a_i(x) = \text{SSIM}(x_0, \tilde{x}_i), \quad \tilde{x}_i = G(\hat{x}_{i-1}, M_{i-1})$$

where $u$ denotes the threshold level for binary classification. After $N$ iterations, we use the $N^{\text{th}}$ SSIM anomaly score $a_N(x)$ for AD evaluation.

### 2.4 MASK INITIALIZATION

We have no mask information at the first iteration step during testing. We use four checkerboard matrices to initialize masks. Figure 6 shows initialized mask examples. A generator encodes pixels of test images in white regions and decodes pixels in black boxes. These black regions are mutually exclusive between four masks and cover target images collectively. Therefore, we combine four generated images into a single whole reconstruction.

### 2.5 ITERATION STEPS AND STOP CRITERIA

We expect no masked region for normal images and some local masked region for anomalous images. Therefore, applying iterative inpainting for some samples on the training dataset could estimate iteration steps enough to reduce masks on normal pixels. Since I3AD decodes only masked regions, $M_{i+1}$ will be almost all subset of $M_i$. Therefore, we could set early stop criteria whether the difference between $M_i$ and $M_{i+1}$ is small against masked regions of $M_i$.

## 3 RESULTS

Table 1: Results for **anomaly detection** on the MVTecAD dataset, expressed in **the AUROC on sample-wise reconstruction errors** for different autoencoders and datasets. We compare the vanilla $L^2$ and SSIM autoencoders(Bergmann et al., 2018) and their iterative projection method (Dehaene et al., 2020) as baselines. Bold font is the best AUC in each category and lightblue background indicates the best method measured by the average AUC on $L^2$ and SSIM anomaly score.

| | | Bergmann et al. (2019) Autoencoders | | | | Dehaene et al. (2020) Iterative projection | | | | Ours Inpainting | |
| | Model | $L^2$AE | | DSAE | | $L^2$AE | | DSAE | | I3AD | |
| | Score | $L^2$ | SSIM | $L^2$ | SSIM | $L^2$ | SSIM | $L^2$ | SSIM | $L^2$ | SSIM |
|---|---|---|---|---|---|---|---|---|---|---|---|
| Textures | **carpet** | 0.476 | 0.531 | 0.401 | 0.538 | 0.447 | 0.513 | 0.402 | 0.449 | 0.602 | **0.605** |
| | **grid** | 0.947 | 0.764 | 0.849 | 0.665 | 0.947 | 0.936 | 0.891 | 0.840 | **0.998** | 0.947 |
| | **leather** | 0.778 | 0.694 | 0.720 | 0.382 | 0.826 | 0.831 | 0.772 | 0.752 | 0.823 | **0.659** |
| | **tile** | 0.763 | 0.706 | 0.533 | 0.380 | 0.810 | 0.815 | 0.684 | 0.662 | 0.978 | **0.983** |
| | **wood** | 0.939 | 0.861 | 0.894 | 0.875 | **0.952** | **0.935** | 0.915 | 0.896 | 0.938 | 0.936 |
| Objects | **bottle** | 0.978 | 0.921 | 0.966 | 0.921 | 0.958 | 0.980 | 0.967 | **0.983** | 0.966 | 0.905 |
| | **cable** | 0.758 | 0.827 | 0.751 | 0.807 | 0.564 | 0.725 | 0.630 | 0.717 | 0.767 | **0.773** |
| | **capsule** | 0.728 | 0.689 | 0.738 | 0.699 | 0.792 | 0.803 | 0.819 | **0.813** | 0.708 | 0.731 |
| | **hazelnut** | 0.902 | 0.841 | 0.898 | 0.845 | 0.956 | 0.942 | **0.957** | 0.912 | 0.930 | 0.895 |
| | **metal nut** | 0.613 | 0.569 | 0.597 | 0.557 | 0.583 | 0.623 | 0.547 | 0.590 | **0.658** | 0.605 |
| | **pill** | 0.825 | **0.838** | 0.780 | 0.818 | 0.741 | 0.746 | 0.721 | 0.725 | 0.783 | 0.780 |
| | **screw** | 0.015 | 0.670 | 0.264 | 0.659 | 0.768 | 0.773 | 0.772 | 0.718 | **0.980** | 0.842 |
| | **toothbrush** | 0.981 | 0.886 | 0.972 | 0.875 | **0.989** | **0.978** | 0.983 | 0.975 | 0.958 | 0.903 |
| | **transistor** | 0.820 | 0.881 | 0.818 | 0.879 | 0.746 | 0.813 | 0.728 | 0.792 | 0.864 | **0.876** |
| | **zipper** | 0.873 | 0.785 | 0.833 | 0.755 | 0.818 | 0.864 | 0.772 | 0.810 | 0.994 | **0.998** |
| Ave. | **Texture** | 0.781 | 0.711 | 0.680 | 0.568 | 0.796 | 0.806 | 0.733 | 0.720 | **0.868** | 0.826 |
| | **Object** | 0.749 | 0.791 | 0.762 | 0.782 | 0.792 | 0.825 | 0.790 | 0.804 | **0.861** | 0.831 |
| | **Total** | 0.760 | 0.764 | 0.734 | 0.710 | 0.793 | 0.818 | 0.771 | 0.776 | **0.863** | 0.829 |

### 3.1 EXPERIMENT CONFIGURATION

We performed experiments with the industrial high-resolution dataset named MVTec Anomaly Detection (MVTecAD) (Bergmann et al., 2019). Please see Appendix B for further details on the dataset. Bergmann et al. (2019) applies different subsampling and evaluations for each model and each category. Dehaene et al. (2020) resizes $128 \times 128$ pixels for objects and extract patches of $128 \times 128$ pixels from resized $512 \times 512$ pixels for textures. In contrast to these previous works, we resize all images to $256 \times 256$ pixels in 15 categories for training and testing, and evaluated all models by same experiment setting.

**I3AD** We train an I3AD generator by 500,000 iterations with 10 images per batch. We use two Adam optimizers (Kingma & Ba, 2014) with learning rates of 0.0001 for the generator and 0.0004 for the discriminator, respectively. Random blushing stroke masks are generated by the algorithm proposed by Yu et al. (2019). Regarding mask generation, we apply 6 by 6 patched four checkerboard matrices masks initialization. We find the threshold hyperparameter $u$ for a generator to achieve low $L_1$ reconstruction errors on training normal images. We apply the I3AD method on samples of training datasets with $u$ set from 0.10 to 0.50 by 0.05. Then, we find a minimal threshold to achieve small $L_1$ reconstruction errors less than the order of 10 after 10 iteration steps. We apply 30 iterations for test images in all categories.

**Baselines** As baseline models, we compare two autoencoders that are trained to minimize different reconstruction loss of $L^2$ and SSIM. Similar to Bergmann et al. (2018), both autoencoder models are parameterized by the same architecture with latent space dimensionality (set to 100) and optimized

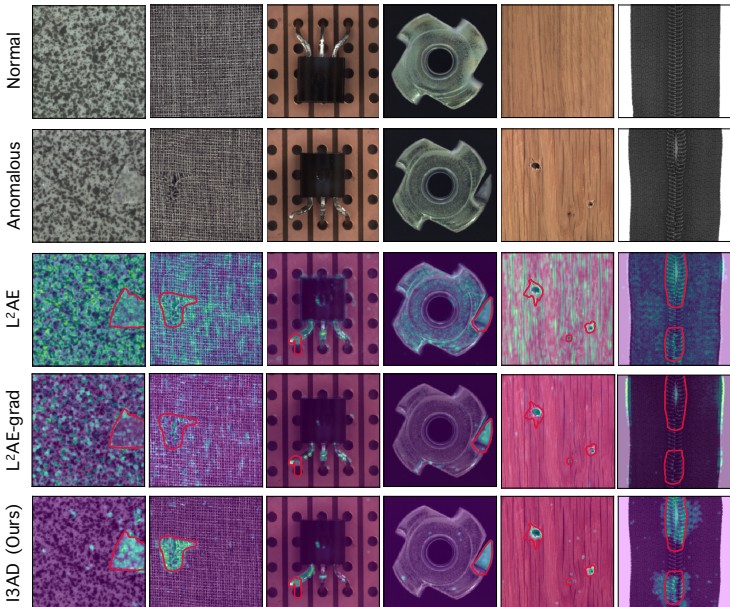

Figure 3: First row: Normal samples of tile, carpet, transistor, screw, wood, and zipper in MVTecAD dataset; Second row: anomalous samples from the aforementioned dataset categories; Third row: Anomaly map by $L^2$ autoencoder (Bergmann et al., 2019); Fourth row: our proposed anomaly map by our I3AD method. Ground truth is represented by red contour, and each estimated anomaly score is highlighted by green.

by Adam with the same learning rate (set to 0.0001). These models are trained by 20,000 iterations on 10 images per batch. For the evaluation of iterative projection method, both $L^2$ and SSIM autoencoders is applied to have additional 50 iterations with step size $\alpha = 0.5$ and $L^1$ regulation weight $\lambda = 0.05$, same parameter setting with Dehaene et al. (2020).

We compute the Area Under the Receiver Operating Characteristics (AUROC) to get a performance independent of the determined threshold for evaluation. The AUROC scores on the sample-wise $L^2$ and SSIM reconstruction errors are computed for anomaly detection, and pixel-wise errors are computed for anomaly localization. For the SSIM Index, their hyperparameters are selected as window size 11, $\alpha = \beta = \gamma = 1$, same with Bergmann et al. (2018). All experiments were performed on a graphics processing unit NVIDIA Quadro RTX 8000 based on a system running Python 3.7, PyTorch library version 1.5, and CUDA 10.1.

## 3.2 QUALITATIVE AND QUANTITATIVE RESULTS

Table 1 shows the AUROC result for **anomaly detection**. Table 5 in Appendix E shows the AUROC result for **anomaly localization**. From both detection and localization, the baselines of vanilla $L^2$ and SSIM autoencoders perform well for object datasets and challenge the high-frequency textures such as carpet, leather, tile, or zipper. Despite different resolution from original experiments, Dehaene et al. (2020)'s approach works and improves AUROC of both localization and detection on many datasets. However, there remains room to improve in the categories where the base autoencoders had a weak result, especially in textures. In anomaly detection, our I3AD outperforms baselines on nine categories and has ave. $+11.1\%$ on textures, ave. $+8.85\%$ on objects and ave. $+13.01\%$ in total from vanilla autoencoders.

Figure 3 shows the qualitative results of a baseline $L^2$AE and our I3AD method. Figure 7 in Appendix G shows the results of the remaining categories. SSIM anomaly score is used for highlighting. Our I3AD has clearly a large improvement in high-frequency categories such as tile, carpet, and zipper.

### 3.3 HYPERPARAMETER SENSITIVITY

In this section, we investigate the hyperparameter sensitivity for the AD results. Compared to typical inpainting tasks to fill various types of images, the inpainting task for AD is relatively easy only to learn every single category. Thus, we skip hyperparameters of the inpainting networks and focus on the hyperparameters regarding mask generation.

**Mask initialization** We changed the patch size on four checkerboard matrices. Figure 4 showed that different patch size initialization has performance differences in the first few steps while converging similar results after enough iterations. We verified that no mask initialization generates anomalous pixels directly and leads to a meaningless result.

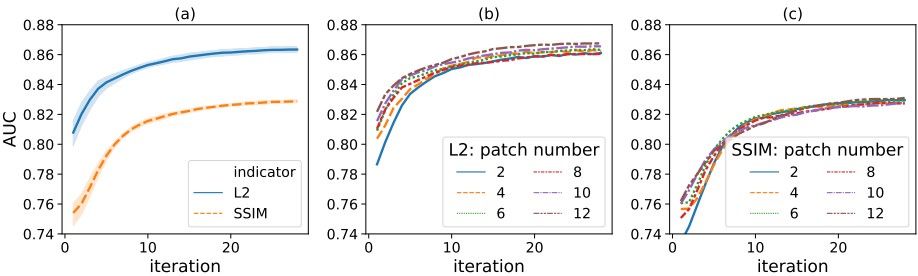

Figure 4: AUC change of different patch-size mask initialization during iteration. Both $L^2$ and SSIM anomaly scores are computed. (a) average AUC result of each anomaly score, (b) AUC result measured by $L^2$ (c) AUC result measured by SSIM

**Mask generation threshold** We determine mask generation threshold $u$ naively from the $L_1$ reconstruction levels on training datasets. Figure 5 shows the AUC score on different threshold selection. Even though the best parameters are unknown and inaccessible from training datasets, we could find a threshold parameter naively. Moreover, since this fitting does not perform equally across data categories, mask generation has room to be optimized in semi-supervised approach.

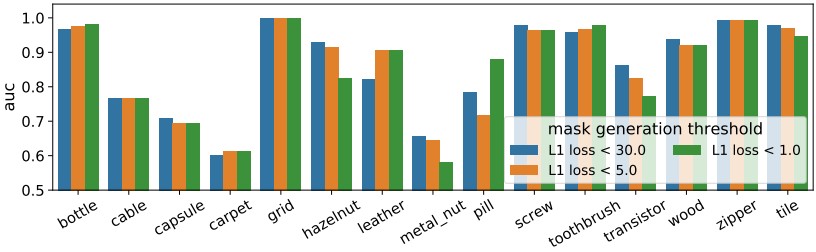

Figure 5: AUC change of different threshold $u$ for mask generation

**Iteration speed** We measure the iteration speed for AnoGAN, Iterative Projection, and our I3AD. Table 2 shows I3AD runs at compatible speed with Iterative Projection method and.

## 4 RELATED WORKS

We see related works about anomaly detection in this section. Appendix H summarizes the architectures of reconstruction-based approach in unsupervised AD. Appendix I shortly summarizes related works about image inpainting.

| models | time per image (seconds) | iterations | iteration | batch calculation |
|---|---|---|---|---|
| AnoGAN | $63.4 \pm 13.4$ | 500 | back-prop | not available |
| Iterative Projection | $0.70 \pm 0.11$ | 50 | back-prop | not available |
| I3AD | $0.58 \pm 0.01$ | 30 | forward | available |

Table 2: Comparison of iteration speed

**Anomaly detection overview**  Bergmann et al. (2019) introduced the MVTecAD dataset and conducted a thorough evaluation of traditional shallow models and recent state-of-the-art deep neural networks for unsupervised AD and segmentation tasks. They showed the evaluated methods do not perform equally across data categories, and there is still room for improvement.

**Autoencoders**  Convolutional autoencoders are commonly used as a base architecture in unsupervised AD. Autoencoders are customized by various reconstruction and regularization losses. Bergmann et al. (2018) applies SSIM Index as reconstruction loss and anomaly map. Zimmerer et al. (2019b) shows that loss gradients are useful anomaly scores to improve the AUC score on unsupervised pixel-wise tumor detection. Zimmerer et al. (2019a) proposes Context-encoding VAE (ceVAE) for unsupervised AD. Context-encoding (Pathak et al., 2016) is a special class of denoising autoencoders (Vincent et al., 2010) to learn inpainting random masks instead of additive Gaussian noise denoising.

Both context-encoding and our I3AD solve inpainting task during training. Motivation and usage are different during testing. Since context-encoding is a data augmentation for feature learning, model architecture is used as vanilla autoencoders. On the other hand, our I3AD uses inpainting masks and encodes the unmasked regions during testing. The iterative process realizes conditional encoding and decoding to estimate unknown masks.

**VAEs**  VAEs is a class of autoencoders to constrain lower-dimensional latent space by Kullback-Leibler divergence, allowing us to conduct anomaly detection by the reconstruction errors and estimated likelihood. On the other hand, Nalisnick et al. (2018); Shafaei et al. (2018); Hendrycks et al. (2018) provide evidence that deep generative models might fail to assign a high likelihood for the out-of-distribution dataset. Several approaches such as training an auxiliary dataset of outliers (Hendrycks et al., 2018), training a background model (Ren et al., 2019), and Likelihood Regret score by fine-tuning test images Xiao et al. (2020), are proposed to mitigate this issue. Since VAEs tend to generate blur images, vanilla autoencoders are popular for local defect detection. Indeed, Matsubara et al. (2018) showed the KL divergence term deteriorates the anomaly sensitivity and proposes to use only reconstruction term for anomaly map. Baur et al. (2018) uses VAEs for unsupervised anomaly segmentation in brain MR scans, and the improvement from autoencoders to VAEs was limited.

**Adversarial training**  Adversarial training is one approach to generate high-resolution images. Schlegl et al. (2017) proposed AnoGAN, which utilizes GANs for unsupervised AD. We do not know a random noise to generate an anomalous-free image close to a targeted image. AnoGAN finds a latent noise sample by iterative update under minimizing the reconstruction errors and the semantic similarity of a discriminator's outputs. Since there was a drawback of long runtime for this calculation, several studies had efforts to reduce its runtime. ADGAN (Deecke et al., 2018) updates the input images and the generator's parameters. Several studies propose to train an encoder network to find an expected random noise from an input image. Their architectures are similar to autoencoders or VAEs with an adversarial term (here referred to as "AEGAN" and "VAEGAN") (Larsen et al., 2016; Dumoulin et al., 2017; Donahue et al., 2017). Their difference is around training processes and loss terms. Efficient-GAN (Zenati et al., 2018) adopted ALI training (Dumoulin et al., 2017) that The discriminator receives the joint pair of latent features and images. GANomaly (Akcay et al., 2018) combines three losses about L1 loss between images, L2 encoder loss between latent features, and adversarial loss. f-AnoGAN (Schlegl et al., 2019) trains an encoder and a generator separately. They examined three architectures regarding loss terms and showed that "izif" architecture best performed of them. Venkataramanan et al. added attention expansion loss for unsupervised and weakly-supervised AD.

**Skip-connection**    Skip-connection is another approach to model small details. U-Net (Ronneberger et al., 2015) is widely applied for cardiac MR, brain tumors, and abdominal CT in supervised segmentation tasks. Akçay et al. (2019); Sabokrou et al. (2018) replace autoencoder by U-Net with AEGAN for unsupervised anomaly detection. Skip-GANomaly (Akçay et al., 2019) inherits GANomaly and uses the same loss terms and anomaly scores. AVID (Sabokrou et al., 2018) is trained as same as vanilla AEGANs. It applies Fully convolutional neural networks (FCNs) for a discriminator to capture the regional information and define each region's regularity likelihood. Oktay et al. (2018) added attention modules into U-Net to be specific to local regions and tested 3D abdominal CT scans.

We tested U-Net architecture for the MVTecAD dataset and verified several models by switching U-Net's four skip-connections on or off. As a result, We faced learning of trivial identity function, and anomalous pixels are reconstructed if models have first or second skip-connection close to input images. I3AD is considered as an application of per-pixel skip-connections that only transport high convinced regions as normal pixels from anomaly score.

**Iterative methods**    In addition to the aforementioned iterative methods such as Likelihood Regret, AnoGAN, and ADGAN, Dehaene et al. (2020) proposed the iterative projection method on trained autoencoders to make blurred images clear at testing time. It iteratively updates an input sample's pixels to minimize reconstruction errors under the constraint on the distance from the original image.

Dehaene et al. (2020)'s method has underlying autoencoders that encode and decode the whole image with back-propagation update. Our I3AD has an underlying of conditional autoencoders with forwarding iterations. Thus, our approach efficiently updates high-resolution pixels and avoids reconstructing complex patterns in background normal pixels.

we summarize where differentiates previous AD methods and our proposed method below.

- Though aforementioned autoencoder-base methods encode and decode whole images, I3AD encodes unmasked regions and decodes masked regions.

- Skip connection passing whole images potentially leads to trivial function. I3AD could learn per-pixel skip-connections where connects local regions the model assigns a high possibility as normal pixels.

- Previous iterative methods used to minimize total reconstruction errors. I3AD targets to minimize the reconstruction errors on normal pixels and maximize ones on anomalous pixels. It attempts to fill in the gap of train and test objectives on unsupervised AD.

- Previous iterative methods require back-propagation with long iteration steps. Ours only uses forward iteration and efficiently updates high-resolution pixels.

## 5    DISCUSSIONS AND FUTURE WORKS

As future works, our I3AD should leverage GAN's unsupervised AD technique. For example, like AVID, a PatchGAN discriminator score could be useful to generate iterative masks. I3AD could be tested on different inpainting models. The coarse-to-fine network with an attention module has a remarkable performance, requiring a large network structure. To reduce the model parameters for speeding up the inference and saving hardware costs, Sagong et al. (2019) proposes weight sharing on coarse and fine generator networks while maintaining performance.

## 6    CONCLUSION

In high-resolution images, autoencoders fail to model small details, which yields blurry image reconstructions. This is especially the case for high-frequency textures, such as carpet, leather, and tile. To tackle this issue, we propose the iterative image inpainting method to reconstruct partial regions adaptively. Our method utilizes the structural similarity measure for inpainting regions, which modifies their structural difference and enhances their reconstruction quality iteration steps. Our method outperforms state-of-the-art results on several categories in MVTecAD datasets and shows an especially large improvement in the texture categories.

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

## A    ANOMALY SCORE OF STRUCTURAL SIMILARITY (SSIM INDEX)

For AD, some per-pixel error measure such as $L^p$-distance is utilized as anomaly score. The structural similarity (SSIM) metric (Wang et al., 2004) could be also employed to capture perceptual similarity(Bergmann et al., 2018; 2019). The SSIM Index defines a structural similarity measure between two $K \times K$ image patches $p$ and $q$, taking into account their similarity in luminance $l(p,q)$, contrast $c(p,q)$, and structure $s(p,q)$:

$$\text{SSIM}(p,q) = l(p,q)^\alpha c(p,q)^\beta s(p,q)^\gamma,$$

where $\alpha, \beta, \gamma \in R$ are hyperparameters. From the patches' mean intensities $\mu_p, \mu_q$, the patches' variances $\sigma_p, \sigma_q$, and their covariance $\sigma_{pq}$, the three measures are defined as

$$l(p,q) = \frac{2\mu_p\mu_q + c_1}{\mu_p^2 + \mu_q^2 + c_1}, \quad c(p,q) = \frac{2\sigma_p\sigma_q + c_2}{\sigma_p^2 + \sigma_q^2 + c_2}, \quad s(p,q) = \frac{2\sigma_{pq} + c_2}{2\sigma_p\sigma_q + c_2}.$$

The constants $c_1$ and $c_2$ ensure numerical stability and are typically set to $c_1 = 0.01$ and $c_2 = 0.03$. By substituting three measures under equally weighting $\alpha = \beta = \gamma = 1$, the SSIM Index is derived by

$$\text{SSIM}(p,q) = \frac{(2\mu_p\mu_q + c_1)(2\sigma_{pq} + c_2)}{(\mu_p^2 + \mu_q^2 + c_1)(\sigma_p^2 + \sigma_q^2 + c_2)}.$$

We have that $\text{SSIM}(p,q) \in [-1, 1]$, and $\text{SSIM}(p,q) = 1$ if and only if $p$ and $q$ are identical.

## B    MVTEC ANOMALY DETECTION DATASET

We performed experiments with the industrial dataset named MVTec Anomaly Detection (MVTecAD) (Bergmann et al., 2019). It contains 5,354 high-resolution color images of ten different objects and five texture categories. All image resolutions are set in the range between $700 \times 700$ and $1024 \times 1024$ pixels. It splits 3,629 images for training and validation and 1,725 images for testing. At test time, there are 467 defect-free images and 1,258 defect images. This dataset includes 73 different defect types, such as defects on the object's surface, structural defects, or the absence of certain parts. These anomalies are manually generated to produce realistic anomalies as they would occur in real-world industrial inspection scenarios. Bergmann et al. (2019) use several resolution for objects and textures on each model. As AnoGAN, both training and testing images are zoomed to $128 \times 128$ pixels for object categories. For textures, $128 \times 128$ patches are extracted from zoomed $512 \times 512$ pixels. As $L_2$ and SSIM autoencoders, they reconstruct patches of $128 \times 128$ pixels for textures and of $256 \times 256$ pixels for objects.

# C ANOMALY LOCALIZATION

Table 3: Results for **anomaly localization** on the MVTecAD dataset, expressed in **the AUROC on pixel-wise reconstruction errors** for different autoencoders and datasets. Same as anomaly detection, we compare the vanilla $L^2$ and SSIM autoencoders(Bergmann et al., 2018) and their iterative projection method (Dehaene et al., 2020) as baselines. Bold font is the best AUC in each category and lightblue background indicates the best method measured by the average AUC on $L^2$ and SSIM anomaly score.

|  | | Bergmann et al. (2019) Autoencoders | | | | Dehaene et al. (2020) Iterative projection | | | | Ours Inpainting | |
|  | Model | $L^2$AE | | DSAE | | $L^2$AE | | DSAE | | I3AD | |
|  | Score | $L^2$ | SSIM | $L^2$ | SSIM | $L^2$ | SSIM | $L^2$ | SSIM | $L^2$ | SSIM |
|---|---|---|---|---|---|---|---|---|---|---|---|
| Textures | carpet | 0.598 | 0.736 | 0.579 | 0.733 | 0.596 | 0.768 | 0.581 | 0.750 | 0.809 | **0.850** |
|  | grid | 0.741 | 0.817 | 0.719 | 0.815 | 0.719 | 0.859 | 0.705 | 0.841 | 0.959 | **0.987** |
|  | leather | 0.739 | 0.758 | 0.775 | 0.730 | 0.735 | 0.812 | 0.779 | 0.923 | 0.833 | **0.938** |
|  | tile | 0.513 | 0.513 | 0.564 | 0.509 | 0.529 | 0.581 | 0.566 | 0.603 | 0.630 | **0.788** |
|  | wood | 0.650 | 0.652 | 0.647 | 0.661 | 0.664 | 0.767 | 0.656 | 0.740 | 0.665 | **0.776** |
| Objects | bottle | 0.886 | 0.924 | 0.883 | 0.931 | 0.884 | 0.930 | 0.875 | 0.926 | 0.923 | **0.950** |
|  | cable | 0.773 | 0.779 | 0.800 | 0.793 | 0.740 | **0.822** | 0.710 | 0.784 | 0.819 | 0.795 |
|  | capsule | 0.839 | **0.919** | 0.815 | 0.898 | 0.843 | 0.914 | 0.819 | 0.909 | 0.733 | 0.854 |
|  | hazelnut | 0.934 | **0.971** | 0.929 | 0.967 | 0.911 | 0.949 | 0.893 | 0.928 | 0.664 | 0.756 |
|  | metal nut | 0.857 | 0.855 | 0.854 | 0.850 | 0.861 | **0.898** | 0.860 | 0.894 | 0.515 | 0.526 |
|  | pill | 0.871 | 0.911 | 0.859 | **0.917** | 0.881 | 0.916 | 0.866 | 0.916 | 0.649 | 0.725 |
|  | screw | 0.824 | 0.974 | 0.870 | 0.974 | 0.834 | 0.975 | 0.877 | 0.975 | 0.852 | 0.959 |
|  | toothbrush | 0.935 | 0.970 | 0.935 | 0.968 | 0.941 | 0.978 | 0.940 | **0.976** | 0.946 | 0.969 |
|  | transistor | 0.817 | 0.867 | 0.802 | 0.862 | 0.831 | 0.875 | 0.852 | **0.889** | 0.596 | 0.651 |
|  | zipper | 0.739 | 0.741 | 0.739 | 0.750 | 0.727 | 0.860 | 0.726 | 0.857 | 0.854 | **0.962** |
| Ave. | textures | 0.648 | 0.695 | 0.657 | 0.690 | 0.649 | 0.757 | 0.657 | 0.771 | 0.779 | **0.868** |
|  | objects | 0.847 | 0.891 | 0.849 | 0.891 | 0.845 | **0.912** | 0.842 | 0.905 | 0.755 | 0.815 |
|  | total | 0.781 | 0.826 | 0.785 | 0.824 | 0.780 | 0.860 | 0.780 | **0.861** | 0.763 | 0.832 |

# D ADDITIONAL COMPARISON BETWEEN GAN MODELS (1)

Table 4: Results for **anomaly detection** on the MVTecAD dataset, expressed in **the AUROC on sample-wise reconstruction errors** for different autoencoders and datasets. We compared AnoGAN (Deecke et al., 2018), f-AnoGAN (Schlegl et al., 2019), and AEGAN that is the base architecture for several models (Zenati et al., 2018; Akcay et al., 2018). Bold font is the best AUC in each category and lightblue background indicates the best method measured by the average AUC on $L^2$ and SSIM anomaly score. We also test $L^2 + L_D$ anomaly scores that combines reconstruction errors and discriminator features as mentioned in Deecke et al. (2018).

| | Model Score | AnoGAN $L^2$ | SSIM | f-AnoGAN $L^2$ | SSIM | $L^2 + L_D$ | AEGAN $L^2$ | SSIM | $L^2 + L_D$ | I3AD $L^2$ | SSIM |
|---|---|---|---|---|---|---|---|---|---|---|---|
| Textures | carpet | 0.360 | 0.399 | 0.311 | 0.512 | 0.433 | 0.488 | 0.565 | 0.132 | 0.602 | **0.605** |
| | grid | 0.661 | 0.517 | 0.451 | 0.733 | 0.236 | 0.937 | 0.826 | 0.942 | **0.998** | 0.947 |
| | leather | 0.393 | 0.298 | 0.385 | 0.271 | 0.288 | 0.648 | 0.290 | 0.624 | **0.823** | 0.659 |
| | tile | 0.627 | 0.476 | 0.724 | 0.514 | 0.735 | 0.540 | 0.410 | 0.591 | 0.978 | **0.983** |
| | wood | 0.820 | 0.779 | **0.940** | 0.849 | 0.905 | 0.933 | 0.875 | 0.930 | 0.938 | 0.936 |
| Objects | bottle | 0.839 | 0.722 | 0.829 | 0.690 | 0.294 | 0.965 | 0.910 | 0.892 | **0.966** | 0.905 |
| | cable | 0.660 | 0.636 | 0.611 | 0.623 | 0.565 | 0.785 | **0.849** | 0.704 | 0.767 | **0.773** |
| | capsule | 0.491 | 0.538 | 0.548 | 0.608 | 0.205 | 0.557 | 0.566 | 0.470 | 0.708 | **0.731** |
| | hazelnut | 0.632 | 0.617 | 0.579 | 0.684 | 0.436 | 0.761 | 0.739 | 0.507 | 0.930 | **0.895** |
| | metal nut | 0.460 | 0.478 | 0.354 | 0.429 | 0.467 | 0.634 | 0.653 | 0.572 | **0.658** | 0.605 |
| | pill | 0.549 | 0.543 | 0.756 | 0.729 | 0.690 | 0.700 | 0.642 | 0.485 | **0.783** | 0.780 |
| | screw | 0.227 | 0.415 | 0.226 | 0.496 | 0.419 | 0.142 | 0.192 | 0.041 | **0.980** | 0.842 |
| | toothbrush | 0.708 | 0.775 | 0.622 | 0.644 | 0.550 | **0.981** | 0.931 | 0.461 | 0.958 | 0.903 |
| | transistor | 0.641 | 0.623 | 0.866 | **0.901** | 0.897 | 0.866 | **0.901** | 0.773 | 0.864 | 0.876 |
| | zipper | 0.587 | 0.506 | 0.695 | 0.566 | 0.291 | 0.901 | 0.780 | 0.875 | 0.994 | **0.998** |
| Ave. | textures | 0.572 | 0.494 | 0.562 | 0.576 | 0.519 | 0.709 | 0.593 | 0.644 | **0.868** | 0.826 |
| | objects | 0.580 | 0.585 | 0.609 | 0.637 | 0.481 | 0.729 | 0.716 | 0.578 | **0.861** | 0.831 |
| | total | 0.577 | 0.555 | 0.593 | 0.617 | 0.494 | 0.722 | 0.675 | 0.600 | **0.863** | 0.829 |

We compared GAN based anomaly detection and localization models. We use same network architecture for our I3AD model and there is only difference that input images are masked or not. AnoGAN takes 500 iterations at test time to find latent random noise. f-AnoGAN has two training steps and an encoder is optimized after training the generator.

# E    ADDITIONAL COMPARISON BETWEEN GAN MODELS (2)

Table 5: Results for **anomaly localization** on the MVTecAD dataset, expressed in **the AUROC on pixel-wise reconstruction errors** for different autoencoders and datasets. We compared AnoGAN (Deecke et al., 2018), f-AnoGAN (Schlegl et al., 2019), and AEGAN that is the base architecture for several models (Zenati et al., 2018; Akcay et al., 2018). Bold font is the best AUC in each category and lightblue background indicates the best method measured by the average AUC on $L^2$ and SSIM anomaly score.

| | Model | AnoGAN | | f-AnoGAN | | AEGAN | | I3AD | |
|---|---|---|---|---|---|---|---|---|---|
| | Score | $L^2$ | SSIM | $L^2$ | SSIM | $L^2$ | SSIM | $L^2$ | SSIM |
| Textures | carpet | 0.553 | 0.576 | 0.487 | 0.504 | 0.571 | 0.695 | 0.809 | **0.850** |
| | grid | 0.537 | 0.548 | 0.561 | 0.532 | 0.777 | 0.863 | 0.959 | **0.987** |
| | leather | 0.624 | 0.712 | 0.557 | 0.721 | 0.798 | 0.754 | 0.833 | **0.938** |
| | tile | 0.544 | 0.508 | 0.536 | 0.513 | 0.539 | 0.495 | 0.630 | **0.788** |
| | wood | 0.598 | 0.629 | 0.657 | 0.655 | 0.627 | 0.617 | 0.665 | **0.776** |
| Objects | bottle | 0.775 | 0.796 | 0.773 | 0.772 | 0.880 | 0.922 | 0.923 | **0.950** |
| | cable | 0.711 | 0.754 | 0.684 | 0.723 | 0.741 | 0.739 | **0.819** | 0.795 |
| | capsule | 0.778 | 0.859 | 0.740 | 0.869 | 0.857 | **0.913** | 0.733 | 0.854 |
| | hazelnut | 0.885 | 0.946 | 0.849 | **0.955** | 0.884 | **0.955** | 0.664 | 0.756 |
| | metal nut | 0.747 | 0.733 | 0.738 | 0.749 | 0.849 | **0.854** | 0.515 | 0.526 |
| | pill | 0.805 | 0.883 | 0.692 | 0.752 | 0.872 | **0.922** | 0.649 | 0.725 |
| | screw | 0.789 | 0.925 | 0.786 | 0.898 | 0.857 | 0.947 | 0.852 | **0.959** |
| | toothbrush | 0.873 | 0.931 | 0.881 | 0.925 | 0.925 | 0.956 | 0.946 | **0.969** |
| | transistor | 0.704 | 0.768 | 0.814 | **0.843** | 0.814 | **0.843** | 0.596 | 0.651 |
| | zipper | 0.710 | 0.734 | 0.702 | 0.727 | 0.734 | 0.743 | 0.854 | **0.962** |
| Ave. | textures | 0.571 | 0.594 | 0.560 | 0.585 | 0.662 | 0.685 | 0.779 | **0.868** |
| | objects | 0.778 | 0.833 | 0.766 | 0.821 | 0.841 | **0.879** | 0.755 | 0.815 |
| | total | 0.709 | 0.753 | 0.697 | 0.742 | 0.782 | 0.815 | 0.763 | **0.832** |

# F    MASK INITIALIZATION

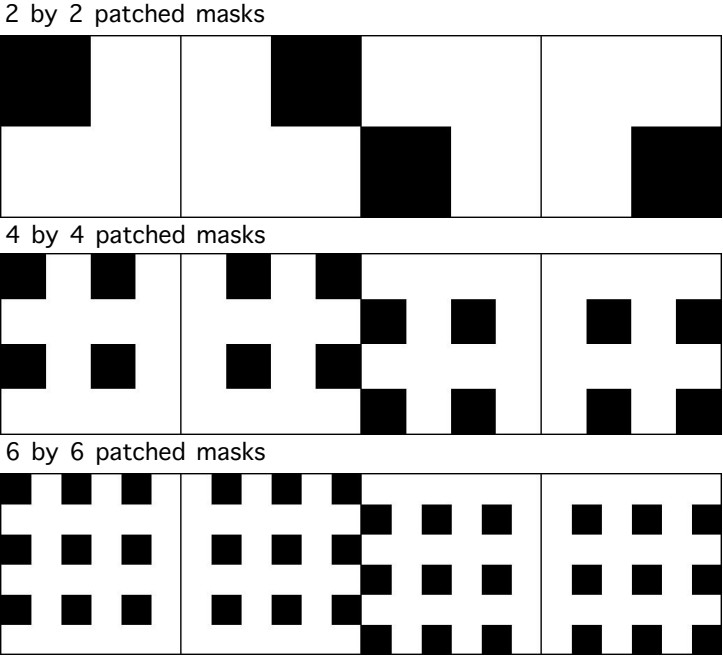

Figure 6: Mask initialization in test iteration step. Images are masked by four masks that includes X by X patched boxes.

A conditional generator in I3AD encodes unmasked pixels and decodes masked pixels. More specially, it decodes whole pixels and uses spatial discounted reconstruction loss to learn to decode only for the masked region. Therefore

as conditional-encoding autoencoder Pathak et al. (2016) During test iterations, the generative model needs the initialized mask at the first iteration step. To cover whole images and segment a few batches, we split images into X by X patches and aggregated them into four masks.

# G QUALITATIVE RESULT

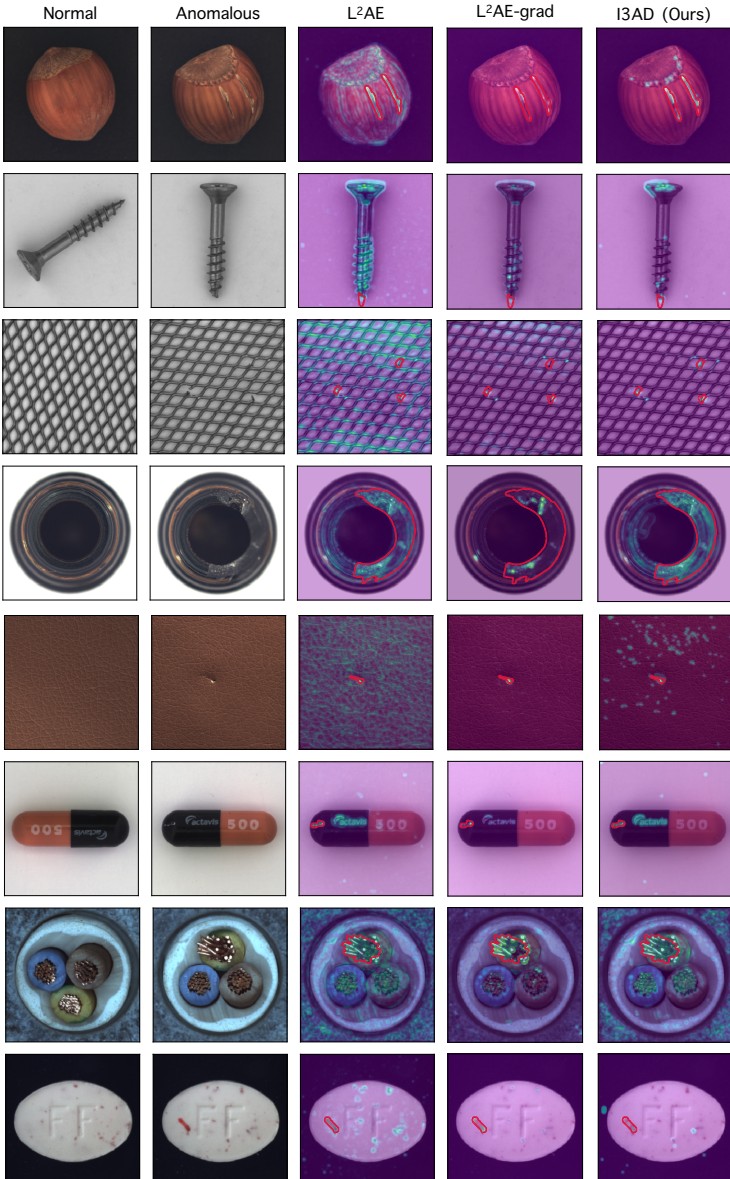

Figure 7: Other qualitative results

# H ARCHITECTURE COMPARISON

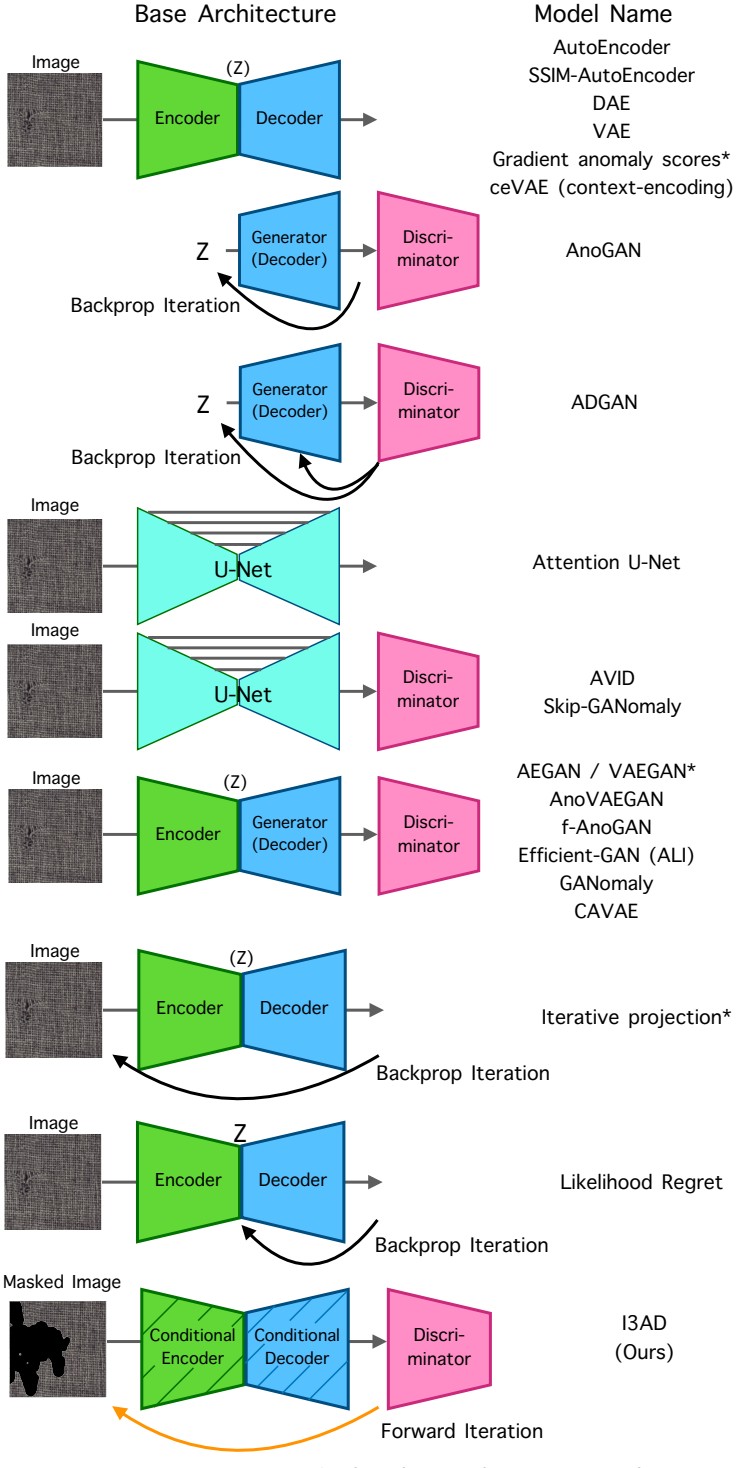

Figure 8: Architecture comparison

We summarize the base model architecture used in recent reconstruction based approach. This is not an all-inclusive list and several studies proposed additional losses and modules with different anomaly score measures in the above base architectures.

# I IMAGE INPAINTING RELATED WORKS

Image inpainting (a.k.a. image completion or image filling) is one of the Image-to-Image translation tasks to cover damaged, deteriorating, or missing parts from remaining parts or alternative contents guided by users. In computer vision, recently, there is much research about the approach based on feed-forward generative models with convolutional networks, especially using conditional generative adversarial networks (cGANs)(Isola et al., 2017).

Iizuka et al. (2017) proposed the model with global and local consistency to handle high-resolution images. To handle irregular masks, Liu et al. (2018) proposed a partial convolution layer that the weights in the convolutional layers are re-normalized by the number of valid pixels from the mask matrix. To produce higher-quality image inpainting, Yu et al. (2018) proposed two-stage architecture with coarse and refinement networks and a contextual attention module in a refinement network. The contextual attention module captures long-range spatial dependencies between masked regions and their surroundings, allowing models to fill large rectangular masks. To handle free-form masks, (Yu et al., 2019) proposed the gated convolution that extends the partial convolution and works pixel normalization and mask updates as trainable weights in neural network layers.

# J ANALOGY OF SSIM-MASK WITH PARTIAL CONVOLUTION LAYER

In inpainting task, the convolutional layer in a generator network assigns the weight balanced on both valid pixels in unmasked region and invalid pixels in masked region. (Liu et al., 2018) proposed partial convolution layer to adapt an irregular shaped mask and re-normalize the dependence of valid pixels. It is computed by the following equations:

$$O = \begin{cases} W * (X \odot \frac{M}{\text{sum}(M)}) & \text{if } \text{sum}(M) > 0 \\ 0 & \text{otherwise,} \end{cases}$$

where let $W$ be the convolution filter weights for the convolution layer, $X$ be the feature values for the sliding window, $M$ be the corresponding binary mask, and $O_{x,y}$ be the output features. The corresponding binary mask $M$ has also the rule-based update in each layer following by

$$M^{l+1} = \begin{cases} 1 & \text{if } \text{sum}(M^l) > 0 \\ 0 & \text{otherwise.} \end{cases}$$

Our I3AD passes through the forward model iteratively, which is interpreted as a very deep generative model. Our SSIM-Mask corresponds to a kind of partial convolutions, which extends to the non-linear structural similarity rule.

