# OpenReview forum: "Iterative Image Inpainting with Structural Similarity Mask for Anomaly Detection"
_ICLR.cc/2021/Conference — Reject_

### Official Review · AnonReviewer2 · 2020-10-26
**An interesting and novel approach to contrastive anomaly detection, but not ready for publication**

**Rating:** 4
**Confidence:** 4

**Review:**

This paper presents a method for contrastive anomaly detection (AD) using an iterative masked conditional autoencoder inpainting approach. An autoencoder network is trained using an adversarial approach to reconstruct a randomly masked part of the input image. At test time a mask is derived from the generated anomaly map (using SSIM index between the input and reconstructed image) and used to mask the input so that the process is repeated N times. The method is shown to produce SOT results on the MVTec AD benchmark.


 PROS:

  * The method is novel and interesting. The iterative masked approach does provide a principled way to increase the signal-to-noise ratio in the reconstructed images.


 CONS:

  * The numbers given in table 1 for the baseline SOT methods are not from the literature. The authors reproduced the numbers but they appear to be much lower than those published. For example, the published value for the average AUC over 15 classes in Bergmann-2019 are L2=0.82;SSIM=0.86, while on table 1, these values are: L2=0.76;SSIM=0.76. This is a substantial difference, which is not explained at all. It makes the proposed method look much better than SOT, while in reality it appears to have similar performance.

  * The number of iterations is not discussed. This is an important hyperparameter as it affects the overall speed of the approach. Only fig 4 shows the progression of the AUC during iterations but no discussion are provided. Also the actual number of iterations used to generate table 1 AUC scores is not given. Assuming the complexity is higher than other SOT methods due to the iterative aspect, and given that the performance is similar to SOT (see above point), this approach now looks a lot less convincing.

  * The paper is not well written and suffers from grammatically wrong and overly complicated sentences, to the point where it becomes distracting and confusing.


 Overall, this is an interesting and novel approach to contrastive anomaly detection. However, I think the paper is not ready for publication. The baseline SOT numbers need to be explained or changed to reflect previously published numbers. The computational complexity of the approach needs to be compared to these baselines. And finally the paper needs to be seriously proof-read.

---

> ### Author Response · Authors · 2020-11-25
> **Additional experiments and comparisons with previous works**
>
> Thank you for your time assessing our paper and your valuable feedback. We revised our papers following your feedback. We appreciate if we can have your review again.
>
> Please let us explain unclear points here.
>
> > The numbers given in table 1 for the baseline SOT methods are not from the literature. The authors reproduced the numbers but they appear to be much lower than those published. For example, the published value for the average AUC over 15 classes in Bergmann-2019 are L2=0.82;SSIM=0.86, while on table 1, these values are: L2=0.76;SSIM=0.76. This is a substantial difference, which is not explained at all. It makes the proposed method look much better than SOT, while in reality it appears to have similar performance.
>
> * We commented table 1 is anomaly detection AUC. We also added anomaly localization AUC in appendix C - E.
>
> >The number of iterations is not discussed. This is an important hyperparameter as it affects the overall speed of the approach. Only fig 4 shows the progression of the AUC during iterations but no discussion are provided. Also the actual number of iterations used to generate table 1 AUC scores is not given. Assuming the complexity is higher than other SOT methods due to the iterative aspect, and given that the performance is similar to SOT (see above point), this approach now looks a lot less convincing.
>
> * we additionally compared recent iterative methods about the number of iterations.
>
> >The paper is not well written and suffers from grammatically wrong and overly complicated sentences, to the point where it becomes distracting and confusing.
>
> * Apology for poor writing. We revised papers and corrected spelling and sentences.

---

### Official Review · AnonReviewer1 · 2020-10-27
**Limited contribution, missing literature and experiments**

**Rating:** 2
**Confidence:** 5

**Review:**

This work proposed a novel learning strategy for unsupervisedly anomaly detection. Particularly, authors propose to use an iterative mask generation process based on image impainting and reduction of a structural similarity metric (SSMI) between the input image and its reconstructed version. For evaluation purposes, authors resort to the public MVTec benchmark, showing better results than the baselines. Please find me comments below:

Strengths
-The paper is generally well written and easy to follow.
-Results show that the proposed method outperforms the baselines.

Weaknesses.
-The methodological contribution is marginal/incremental. Similarly to several methods in weakly supervised segmentation (see [1] for example) authors use iterative steps to refine the initial segmentation mask. The only difference is that instead of mine regions based on classification activation maps and classification scores the authors use a structural similarity pixel-wise metric. Beyond that there is nothing novel in the proposed methodology.

-Some ideas/motivations are unclear. For example, I don’t really understand why the mask initialization is needed at test time. Further, authors also mention in the Appendix. D that to cover whole images, they are split into X by X patches and then aggregated into four masks. Please provide more details since this is unclear.

-Literature is poorly conducted with many relevant recent papers missing (furthermore, the literature comes late in the paper). In addition to the previous paper in weakly supervised segmentation, the works in [2-6] are recent works in anomaly detection, just to name a few. Authors should include all this papers, discuss their limitations and show how the proposed work can overcome these drawbacks. Authors mention that auto-encoders produce blurry images, which is true. Nevertheless, works including an adversarial discriminator have somehow addressed the issue of blurred reconstructed images. Given all this, authors should better motivate their work.

-Among previous missing papers, there is the work in [2], which also employs an impainting strategy coupled with an adversarial model. Which are the differences with respect to this work?

-Experiments need to be significantly extended. First, authors merely include two methods in their evaluation, while there exist more than those used in the comparisons. This is particularly important since some recent methods which have been omitted in the literature review made by the authors (e.g., [4]) significantly outperform the proposed method (0.863 vs 0.90 in AUC). Second, authors report results in terms of AUC, while I strongly suggest that they use the accuracy for individual classes, and the AUC as average of the classes. The reason for this is to better compare to related work (See for example Table 6 in [4]).


References:
[1] Wang et al. Weakly-supervised semantic segmentation by iteratively mining common object features. CVPR 2018.
[2] Sabokrou et al. Avid: Adversarial visual irregularity detection. ACCV 2018
[3] Perera et al. Ocgan: One-class novelty detection using gans with constrained latent representations. CVPR 2019.
[4] Venkataramananet al. Attention Guided Anomaly Localization in Images. ECCV 2020.
[5] Deecke et al. Image anomaly detection with generative adversarial networks. In Joint european conference on machine learning and knowledge discovery in databases 2018.
[6] Li et al. Exploring deep anomaly detection methods based on capsule net. In Canadian Conference on Artificial Intelligence 2020


Minor: The paper in David Dehaene, Oriel Frigo, S´ebastien Combrexelle, and Pierre Eline. Iterative energy-based projection on a normal data manifold for anomaly localization. arXiv preprint arXiv:2002.03734 is an ICML 2020 published paper,

---

> ### Author Response · Authors · 2020-11-25
> **Additional experiments and comparisons with previous works**
>
> Thank you for your time assessing our paper and your valuable feedback.
> We revised our papers following your feedback. We appreciate if we can have your review again.
>
> Please let us explain unclear points here.
>
> > Some ideas/motivations are unclear. For example, I don’t really understand why the mask initialization is needed at test time.
> - Your point is essential why we need the mask initialization. Indeed, Patheck [1] use inpainting task at training as one of the denoising applications. They apply vanilla autoencoder at testing without any masks.
> - We tested no mask input at first iteration step. This is not trivial identity function but reconstructs anomalous regions directly.
> - We motivate a generator works as same role at training and testing. A generator encodes only normal pixels and decodes masked regions even if we don't know there are anomalous or normal pixels. we don't have the guarantee the reconstruction behavior when local defects is encoded.
>
> > Further, authors also mention in the Appendix. D that to cover whole images, they are split into X by X patches and then aggregated into four masks. Please provide more details since this is unclear.
> - Black boxed region is masked and white region is unmasked. A generator encodes pixels in white region and decodes only black boxed region.
> - Since these black boxes in 4 checkboard masks are not overlapped, we use 4 copies of image and combine their reconstructed regions.
>
> > The methodological contribution is marginal/incremental.
> > Literature is poorly conducted with many relevant recent papers missing
>
> - Apology for weak summarization about recent works. We should motivate literature and investigate what are previous solutions and where we could add values for them.
> - We largely rewrote Related Works and summarized architecture of recent unsupervised AD methods in Appendix H.
>
> • To my best knowledge, previous unsupervised AD methods are based on autoencoders. We propose conditional autoencoder (conditional GAN) at testing.
> • Previous methods used to minimize total reconstruction errors. Our I3AD targets to minimize the reconstruction errors on normal pixels and maximize ones on anomalous pixels. It attempts to fill in the objective gap of train and test objectives on unsupervised AD.
>
> Thanks for listing papers
>
> [1] targets object localization from image tag information. They convert image tag scalar information to segmentation prior via super-pixels. Their iterative process is image tag conversion network and an segmentation network.
>
> > Among previous missing papers, there is the work in [2], which also employs an impainting strategy coupled with an adversarial model. Which are the differences with respect to this work?
>
> [2] (AVID) consists of U-Net + adversarial term. Though it names "inpainting network", this is same with other reconstruction-base unsupervised AD approach.
>
> As [4], they use weakly-supervised information and pre-trained model and not easy to compare, we verified their base architecture of AEGAN.
>
> We tested [5] AnoGAN in our experimental setting and added result in Appendix D and E.
>
> > Experiments need to be significantly extended.
> * We added additional comparisons on anomaly detection and localization tasks.
> * For both tasks, we compute AUC as average of the classes.
> * Since IoU and accuracy depends on the threshold of anomaly scores in each model. To compare several models, we keep AUC.

---

### Official Review · AnonReviewer4 · 2020-10-29
**This work, Iterative Image Inpainting for Anomaly Detection (I3AD), has focused on the reconstruction-based unsupervised Anomaly Detection (AD) which is known as rare event detection task.**

**Rating:** 6
**Confidence:** 1

**Review:**

 This approach relies on learning merely normal training sample and then
distinguishing between normal and abnormal events regarding the threshold of the reconstruction error.
I3AD exploits a combination of per-pixel identity function and conditional auto-encoder which is
capable of encoding only normal regions and decoding just anomalous regions.
Weaknesses:
-Although this paper has attempted to propose an efficient algorithm for anomaly detection, the
the general idea of this paper is not sufficiently innovative.
-The most important works on the topics are not cited.
-The proposed method is not comprehensively compared with the other solution.
-A comprehensive analysis of the complexity of the proposed method is required. It seems the proposed method achieves the state-of-the-art performed at the expense of complexity.

Strengths:
This paper is well written so that the structure is easy to follow. The majority of the obtained results
are better than the state-of-the-art has been reported in this paper. The proposed method has tried
to tackle two noticeable issues of using auto-encoders for the task of anomaly detection including
their defect inappropriately modeling small detail as well as the existing object mismatching that
the models are trained to minimize total reconstruction errors

---

> ### Author Response · Authors · 2020-11-25
> **Thank you for your time assessing our paper**
>
> Thank you for your time assessing our paper. We revised our paper following AnonReviewers' comments. We appreciate if we can have your review on revised version.
> Best regards,

---

### Official Review · AnonReviewer3 · 2020-10-30
**Simple and interesting idea with weak experimental evaluations**

**Rating:** 5
**Confidence:** 3

**Review:**

This paper presents an impainting-based method for anomaly localization on images. In the training time, a conditional GAN-based generative modeling approach is adopted. In the test time,  a mask matrix is adaptively estimated by thresholding the structural similarity index measure (SSIM) between the original images and reconstructed images. The idea is very intuitive and experiments demonstrate improved performance (especially on textures) over two recent baselines methods.

Strong points:

- This paper improves the test time in unsupervised anomaly detection with an iteration scheme. Although similar ideas of iteratively focusing on refining the low-error regions and leaving alone the high-error regions have been proposed before, the residual-thresholding scheme proposed in this paper looks even simpler.

- Another novelty is in the training time, the proposed approach using a conditional and GAN-based approach. This is in contrast with autoencoder-based reconstruction approaches which may lead to blurry image reconstruction.

Weak points:

-  The experiment evaluation is not quite convincing. Some of these numbers look worse than what is reported by Bergmann et al., 2018 and Dehaene et al., 2020. Are there any settings different when reproducing the baseline results?

- The qualitative comparison with Dehaene et al. 2020 is missing in Figure 3.

- Figure 4 clearly shows the AUC improves with iterations. Would this SSIM-threshold scheme also benefit other approaches (e.g., a conditional autoencoder)? Are these performance improvements over baselines mainly attributed to the choices of free-form random mask and GAN, or the proposed SSIM-threshold scheme? Would it be better than other iterative schemes such as iterative projection with the same trained model? More ablation studies would be helpful.

Other points to clarify:

- I guess Table 1 AUROC is for pixel-wise segmentation rather than image classification, right?

- How does the checkboard mask initialize the test? Applying different masks to 4 copies of images and then merge?

- Stopping rule of iteration

Overall, I think the idea in this paper is very interesting. I still have some conservations about experimental evaluation (as explained above) against me voting for acceptance.

---

> ### Author Response · Authors · 2020-11-25
> **Explanation of comparison with previous works - different experimental setting and statistics**
>
> Thank you for your time assessing our paper and your valuable feedback.
> We revised our papers following your feedback. We appreciate if we can have your review again.
>
> Please let us explain unclear points here.
>
> > The experiment evaluation is not quite convincing. Some of these numbers look worse than what is reported by Bergmann et al., 2018 and Dehaene et al., 2020. Are there any settings different when reproducing the baseline results?. I guess Table 1 AUROC is for pixel-wise segmentation rather than image classification, right?
>
> - Our Table 1 is the AUROC on sample-wise reconstruction errors for anomaly detection. That is why the number is smaller than reported by Bergmann et al., 2018 and Dehaene et al., 2020. They reported the AUROC on pixel-wise reconstruction errors for anomaly localization.
> - We also added the AUROC on pixel-wise reconstruction errors for anomaly localization in Appendix C.
> - We apply different experimental setting about image resolution and evaluation. Original MVTecAD image resolution are ranged between 700 x 700 and 1024 x 1024 pixels. Bergmann et. al. patches of 128 x 128 pixels for textures and ones of 256 x 256 pixels for objects. They reconstruct patches at a stride of 30 pixels and average the resulting anomaly maps. Dehaene et al., 2020 resizes 512 x 512  pixels and crop patches of 128 x 128 pixels for textures, and resize 256 x 256 pixels for objects. We simply resize 256 x 256 pixels for all categories in training and testing.
>
> > The qualitative comparison with Dehaene et al. 2020 is missing in Figure 3.
> - We added qualitative comparison with Dehaene et al. 2020. Thank you.
>
> > Figure 4 clearly shows the AUC improves with iterations. Would this SSIM-threshold scheme also benefit other approaches (e.g., a conditional autoencoder)? Are these performance improvements over baselines mainly attributed to the choices of free-form random mask and GAN, or the proposed SSIM-threshold scheme? Would it be better than other iterative schemes such as iterative projection with the same trained model?
>
> - We summarized recent anomaly detection models in Appendix H. Previously they studied autoencoders and a conditional autoencoder has room to study.
> - To apply a conditional information and estimate unknown masks, iterative scheme is utilized here. So we think we could customize mask matrix or anomaly maps. As AnonReviewer1 commented, other GAN anomaly detection such as AVID takes discriminator features for anomaly map.
> - Instead of SSIM-Mask, we tried to use MSE. However, it couldn't cover and mask the anomalous defect well.
>
> >How does the checkboard mask initialize the test? Applying different masks to 4 copies of images and then merge?**
> - Yes, we merge 4 copies of images. these black boxes in 4 checkboard masks are not overlapped, we just combine reconstructed regions.
>
> >Stopping rule of iteration
> - We added stopping rule shortly in revised version. Since our iteration method don't train any parameters, they easily stops when there is no difference on mask(t-1) and mask(t).
>
> Best,

---

### Decision · Program_Chairs · 2021-01-07
**Final Decision**

**Decision:**

Reject

**Comment:**

The initial reviews were a bit split. R4 was slightly positive, R3 was slightly negative, and both R1 and R2 voted for rejection. The main issue was lack of proper comparisons with the SOTA methods and missing references. In the rebuttal, the authors added additional experiments as requested, but R1 and R2 were not convinced by the new results. In particular, R1 pointed out that even the unsupervised setup in [4] achieved 0.89 AUC, outperforming 0.86 as reported in the paper. The AC agrees with R1 and R2 that the paper cannot be published until more thorough comparisons are conducted.